# Environmental Impact of Subsidy Concepts for Stimulating Car Sales in Germany

**Malte Scharf** [ID]**, Ludger Heide \*** [ID]**, Alexander Grahle** [ID]**, Anne Magdalene Syré**
**and Dietmar Göhlich** [ID]

Department of Methods for Product Development and Mechatronics, Technical University of Berlin,
10623 Berlin, Germany; malte.scharf@campus.tu-berlin.de (M.S.); alexander.grahle@tu-berlin.de (A.G.);
a.syre@tu-berlin.de (A.M.S.); dietmar.goehlich@tu-berlin.de (D.G.)
**\*** Correspondence: ludger.heide@tu-berlin.de; Tel.: +49-(0)30-314-73-858

**Abstract:** In 2020, vehicle sales decreased dramatically due to the COVID-19 pandemic. Therefore, several voices have demanded a vehicle subsidy similar to the "environmental subsidy" in Germany in 2009. The ecological efficiency of vehicle subsidies is controversially discussed. This paper establishes a prognosis of the long-term environmental impacts of various car subsidy concepts. The $CO_2$ emissions of the German car fleet impacted by the purchase subsidies are determined. A balance model of the $CO_2$ emissions of the whole car life cycle is developed. The implementation of different subsidy scenarios directly affects the forecasted composition of the vehicle population and, therefore, the resulting life-cycle assessment. All scenarios compensate the additional emissions required by the production pull-in within the considered period and, hence, reduce the accumulated $CO_2$ emissions until 2030. In the time period 2019–2030 and for a total number of 0.72 million subsidized vehicles—compensating the decrease due to the COVID-19 pandemic—savings of between 1.31 and 7.56 million t $CO_2$ eq. are generated compared to the scenario without a subsidy. The exclusive funding of battery electric vehicles (BEVs) is most effective, with an ecological break-even in 2025.

**Keywords:** subsidy; automotive industry; prognosis; COVID-19; environmental impact; life-cycle analysis

---

## 1. Introduction

As a result of the containment measures against the COVID-19 pandemic, international vehicle sales collapsed dramatically in the first half of 2020. The German Association of the Automotive Industry (Verband der Automobilindustrie, VDA) predicts a decline of −23% of new passenger car registrations in Germany compared to the prior year [1]. With more than 800,000 employees and an annual turnover of 435 billion EUR in 2019, the German automotive industry is essential for prosperity and employment in Germany [2]. According to the VDA president Hildegard Müller, the massively reduced production will lead to a decrease in employment [1].

From an economic point of view, the COVID-19 pandemic shows some similarities to the 2008 financial crisis: At that time, the passenger car registrations in Germany decreased to the lowest level since the German reunification. For the following year, the forecast without any car sales stimulation predicted 2.8 million new registrations, almost 0.3 million less than the already historically low number of 3.09 million new registrations in 2008.

In reaction, the German government decided to introduce a subsidy program in 2009, in which a purchase bonus for new cars could be earned if the old ones were scrapped. One target of the "environmental bonus" was to replace old cars with high specific emissions with new and more efficient ones. The government's goal was to reduce pollution and stimulate car sales at the same time.

Thus, the number of new registrations in 2009 rose by 23.3% to 3.81 million vehicles and provided, according to Höpfner et al. [3], reduced pollution due to the rejuvenation of the German passenger car fleet.

Due to the renewed economic challenges as a result of the COVID-19 pandemic, a considerable number of voices from the automotive industry [4], automotive lobby [5], and parts of German politics [6] demand a subsidy concept similar to the "environmental bonus" introduced in 2009. The scope of this potential subsidy has not yet been worked out. Unlike in 2009, electromobility has found its way into the automotive market, and therefore, specific funding of battery electric vehicles (BEVs) and plug-in hybrid electric vehicles (PHEVs) is conceivable to further reduce German car fleet emissions. Contrary to the concept proposed in this paper, existing subsidy concepts for electric and hybrid cars [7] do not require a replacement of conventional cars. An additional promising concept includes rewarding purchases of smaller vehicles.

The goal of this paper is a quantified evaluation of various subsidy concepts with regards to the ecological aspects, focusing on the $CO_2$ emissions. To provide a holistic view on the environmental impact of the various subsidy concepts, the cradle-to-grave life cycle, including production, operation, and end-of-life (EoL) emissions, is considered.

## 2. Literature Review

As presented in Section 1, the financial crisis of 2008 and the introduced car subsidy show some similarities to the current situation due to the COVID-19 pandemic.

The "environmental bonus" introduced in Germany has been analyzed extensively. Shortly after the "environmental bonus" was introduced, Höpfner et al. [3] determined a positive ecological effect of the subsidy. In addition, they demonstrated that the $CO_2$ emissions of the pulled-forward manufacturing were compensated after 6000 km driving distance due to the reduced use-phase emissions of the new cars.

Klößner et al. [8] examined the impact of European car scrappage programs on new vehicle registrations and respective $CO_2$ emissions. Using a multivariate synthetic control method with time series of economic predictors, they found that the German subsidy had a positive effect on stabilizing the car market. However, the economic benefit caused 2.4 million tons of additional $CO_2$ emissions according to their work.

Various economic examinations of the "environmental bonus" have been made. The micro- and macroeconomic effects of the "environmental bonus" were investigated by Läufer et al. [9]. They concluded that the "environmental bonus" was not very effective, but created macroeconomic stability.

Müller et al. [10] analyzed the impact of the subsidy on the overall car sales by using a dataset provided by the Organisation for Economic Co-operation and Development (OECD) for 23 countries and found a positive effect of car scrappage programs on overall car sales as long as the subsidy was in place.

The pull-forward effect of the German car scrappage scheme was examined by Böckers et al. [11], who created a monthly dataset of new car registrations owned by private consumers. According to them, the small and upper small car segments benefited specifically from the scrappage program, as they made up 84% of the newly registered cars during the program.

In response to the financial crisis, other countries besides Germany introduced car subsidy programs. The "Summary of the Consumer Assistance to Recycle and Save Act of 2009" (CARS) was launched in 2009 by the US government [12] and has been broadly reviewed. Lenski et al. [13] analyzed the net effect of CARS on greenhouse gas emissions from a full vehicle-life-cycle perspective. They found that CARS had a one-time effect of preventing 4.4 million metric tons of $CO_2$ eq. emissions, about 0.4% of US annual light-duty vehicle emissions.

By comparing the predicted fuel economy without the existence of the program and the actual data, Sivak et al. [14] determined an improved average fuel economy of the US passenger car fleet in July and August 2009.

Li et al. [15] investigated the effects of the CARS program on new vehicle sales and the environment. By using Canada as the control group in a difference-in-differences framework, they determined that CARS increased new vehicle sales only by about 0.37 million during July and August of 2009, implying that approximately 45% of the spending went to consumers who would have purchased a new vehicle anyway. They calculated a reduction of the $CO_2$ emissions by 9–28.2 million tons.

In summary, the literature shows disagreement about the environmental impact of the car subsidy in 2009, and does not answer whether a new subsidy would have a positive impact on the environment.

To the best of our knowledge, there is no study that predicts the environmental impact of a car subsidy that addresses the decrease of vehicles sales due to the COVID-19 pandemic. We perform a life-cycle analysis of the German passenger fleet with a level of detail that no previous study has shown.

## 3. Methodology

The model developed in this paper calculates the annual $CO_2$ emissions of the German passenger car fleet iteratively starting in 2019. New registrations and decommissions of the present year lead to the fleet data of the subsequent year. For the baseline of the $CO_2$ emissions, the impact of the COVID-19 pandemic on the car sales profile is considered. Three different subsidy concepts are applied to the baseline scenario, and variations of the $CO_2$ emissions are determined.

### 3.1. Scope

To analyze the environmental impacts of various subsidy concepts, the full life cycles of passenger cars are taken into account. Therefore, temporal and geographical system boundaries must be defined. To predict the long-term impacts, the time period covers the years from 2019 to 2030. Due to the rapid technological progress, the reliability of the predicted data after 2030 decreases significantly.

As effects that extend beyond the national borders are subject to large uncertainties, the system boundary is drawn around Germany. After deregistration from the German transport system, only the EoL (decommissioning) emissions and no further driving performances will be accounted for in the life-cycle analysis, as shown in Figure 1.

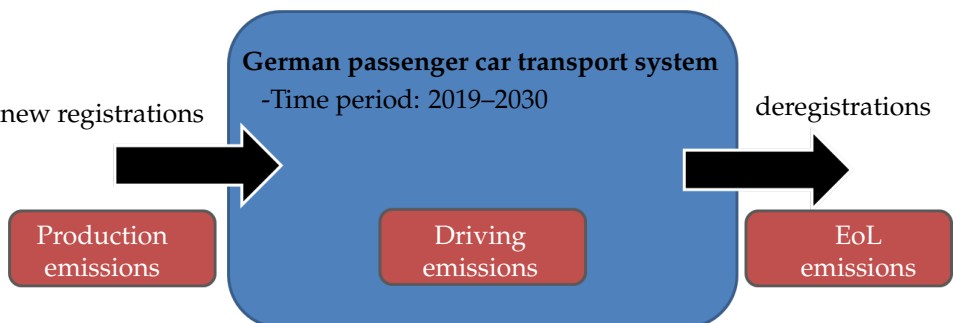

**Figure 1.** System boundaries.

The car population is categorized according to two main attributes: segment and drive train technology. The segment categorization is based, amongst other parameters, on the weight and size of the vehicle, according to the Federal Motor Transport Authority (Kraftfahrtbundesamt, KBA) clustering method [16]. In this work, we use an English translation of the segment names as used by the KBA (Table 1).

**Table 1.** Segment translations.

| Segment Name According to the KBA [16] | English Segment Name |
|:---:|:---:|
| Minis | mini class |
| Kleinwagen | small class |
| Kompaktklasse | compact class |
| Mittelklasse | middle class |
| Obere Mittelklasse | upper middle class |
| Oberklasse | upper class |
| Sport Utility Vehicles (SUVs) | sport utility vehicles (SUVs) |
| Mini-Vans | mini vans |
| Großraum-Vans | large vans |
| Sportwagen | sports cars |
| Geländewagen | off-roaders |
| Wohnmobile | caravans |

In this analysis, there is no consideration of the segments of sports cars and caravans. This is because of their exclusive use for recreational or leisure purposes, which makes the expenditure of public funds for these vehicles unjustifiable. Furthermore, commercial vehicles like trucks are excluded, since the subsidy presented here is focused solely on private customers. Finally, off-roaders are excluded, as they are used either commercially (e.g., forestry) or for leisure purposes. This limits the scope to 84% of the whole German passenger car population.

Furthermore, we focus only on the following drive train technologies: gasoline, diesel, electric, and plug-in hybrid. In this context, a plug-in hybrid vehicle contains a gasoline engine and an electric drive train.

### 3.2. Current Vehicle Distribution

At first, a model was defined that describes the German passenger car population for the next ten years. Therefore, 2019 is considered as a baseline with real data. From 2020–2030, we take recourse to prognosis data, as shown in Section 3.3.

To obtain a sufficient dataset for the 2019 baseline, the vehicle data from the 2017 "Mobility in Germany" (Mobilität in Deutschland, MID) study by the German Federal Ministry of Transport and Digital Infrastructure [17] were categorized into segments and drive train technologies according to Section 3.1. Given the negligible change of 0.2 years in the average vehicle age between 2017 and 2019 [18], we assumed that the overall age distribution of the German vehicle stock did not change significantly. Therefore, a linear transformation of the 2017 data was made to obtain the age distribution of the segments in 2019. Figure 2 presents the resulting age distribution of the German passenger car fleet.

The same data are also available for the vehicle drive train technology and are linked with the year of manufacture and the segment category.

Combined with the number of the total vehicle stock [19], the baseline for 2019 was built.

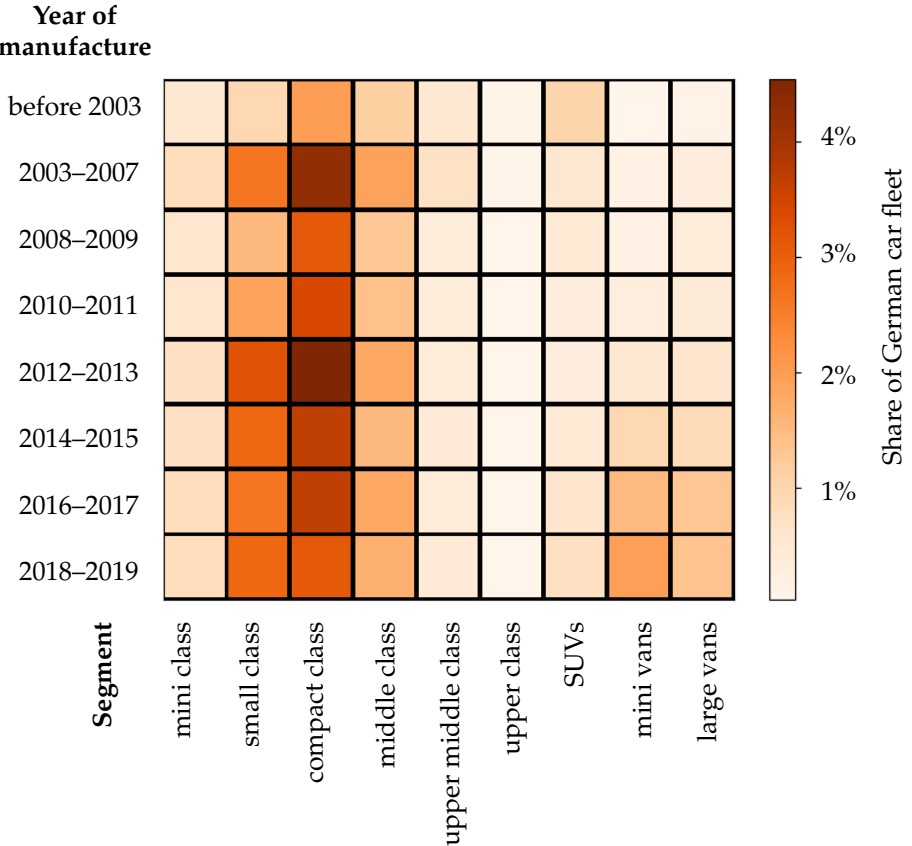

**Figure 2.** Age distribution of the German vehicle fleet in 2019 (vehicles in scope: 82% of overall vehicle stock).

*3.3. German Car Population Development until 2030*

In order to predict the car population until 2030, estimations for new registrations and deregistrations must be made.

3.3.1. New Passenger Car Registrations

To determine new passenger car registrations, the total number and their distribution in the categories defined in Section 3.1 are needed. The annual number of new registrations and their classification regarding the drive train technology were obtained from the "proKlima" scenario of Agora 2018 [20] (Figure 3).

Additionally, an estimation concerning the segments was created based on the trend of the new registration numbers from previous years. For the large van, mini van, upper class, and upper middle class segments, we assumed that the share of new registrations remains constant. In addition, we assumed that the new registration share of segments not considered in this analysis will stay constant, which results in a non-varying share of the overall scope of 82%. A noticeable growth of the sport utility vehicle (SUV) segment has been observed over the last years [21]. The vehicle segment development is visualized in Figure 4.

This trend is expected to continue and is anticipated to reach an annual new registration share of 40% in 2030. Since this value is based on an assumption, a sensitivity analysis is performed in Section 4.4.2.

The growth in the SUV segment was subtracted proportionally from the other segments depending on their share in the reference year 2019. The shares of the large van and mini class segments were not modified, as customers are unlikely to change to an SUV due to the completely different usage profiles.

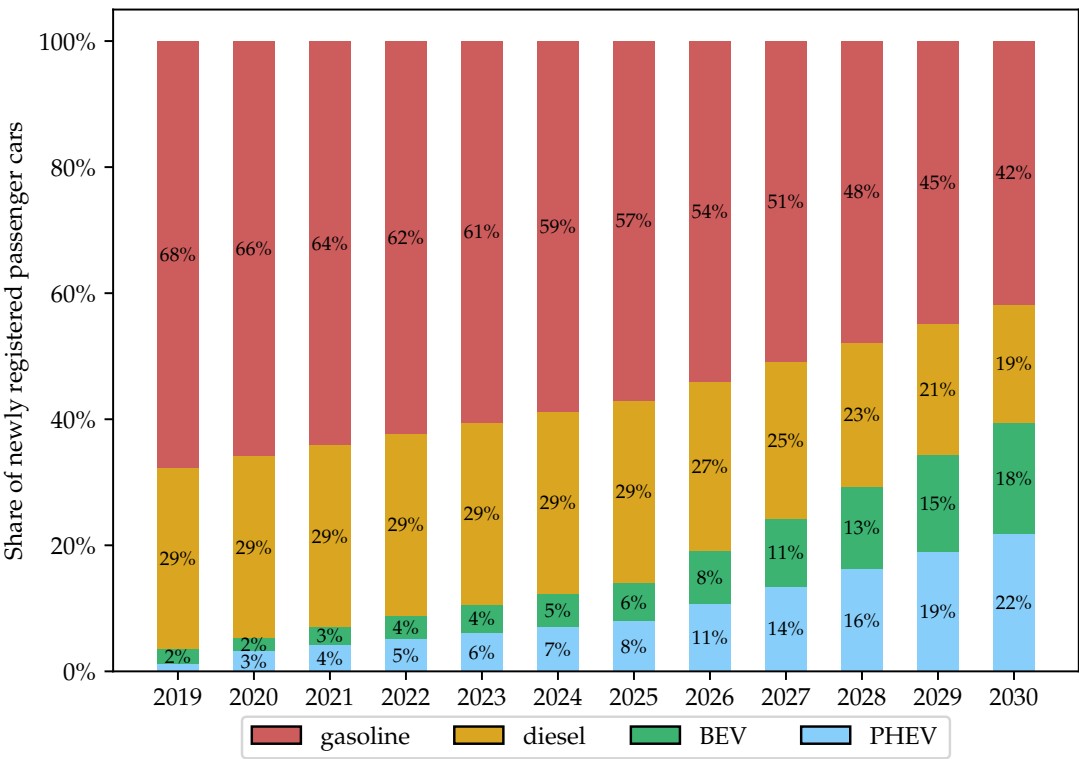

**Figure 3.** Linear interpolation of the "proKlima" drive drain technology scenario.

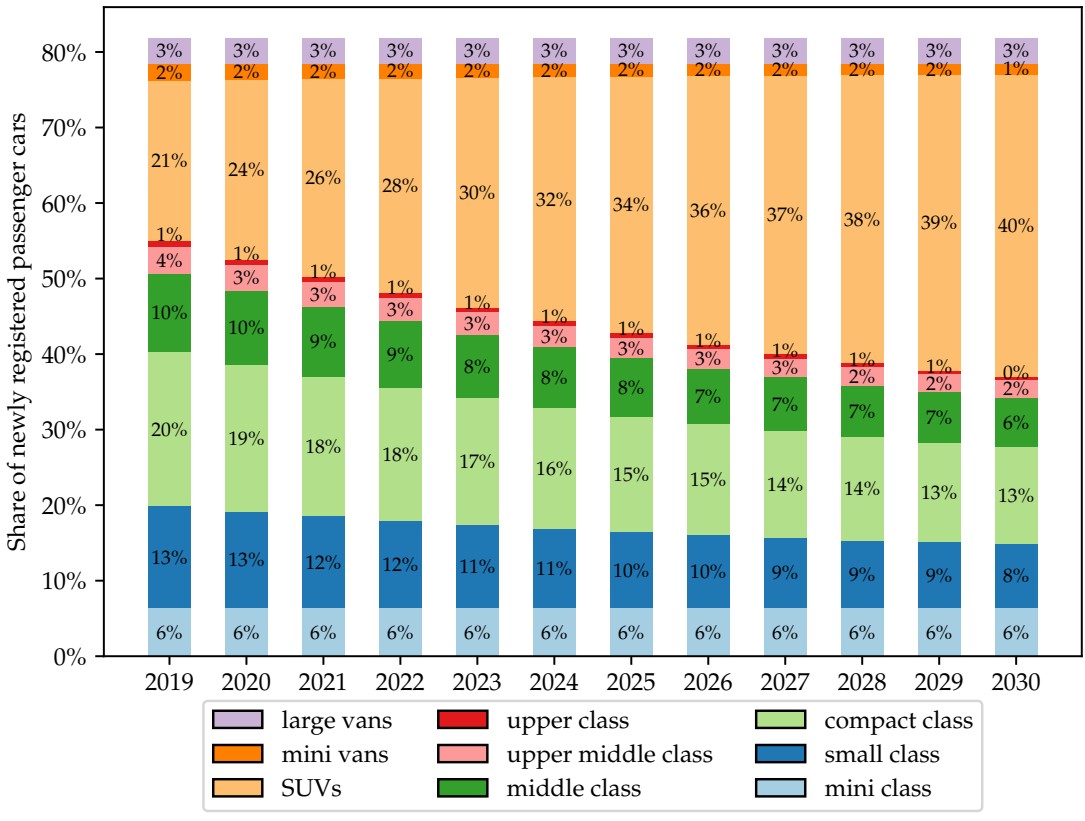

**Figure 4.** Segment scenario.

### 3.3.2. Vehicle Disposal

In this analysis, we assumed that every vehicle that is deregistered from the German transport system, as defined in Section 3.1, will immediately be disposed of, meaning scrapped or recycled. As usage profiles of exported vehicles are unknown, the vehicles are disposed of simultaneously with their deregistration in this model.

For the last four years, a consistent age-dependent deregistration trend can be seen in Figure 5. Therefore, the future deregistration rates are assumed to be identical to those in 2019. The peak after three years can be explained by the end of many leasing contracts, which can contribute to exportation of vehicles. Additionally, the first legally required technical inspection ("Hauptuntersuchung", HU) of the vehicles and the associated phasing out of early defective vehicles take place in this period. This might add to the observed peak. The peaks that occur every two years, especially for older vehicles, are also due to the interval of the mandatory technical inspections in Germany. As technical defects are detected and repairs may become necessary, it is often cheaper to scrap or export the vehicles than to continue their operation.

As explained in Section 3.4, vehicle production is modeled separately from deregistrations, and only for new vehicles. The negligible number of new registrations (negative deregistration) for very old vehicles (probably as vintage vehicles) is ignored.

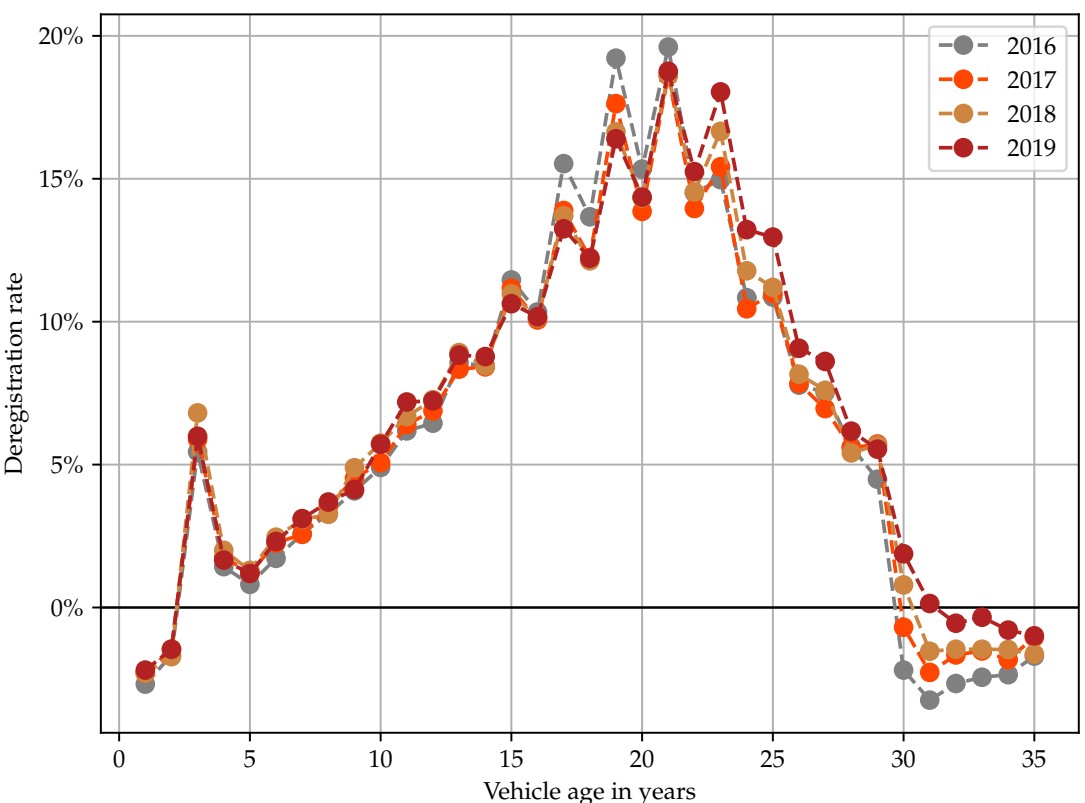

**Figure 5.** Age-dependent deregistration data for 2016–2019 (based on [22]).

### 3.4. Modeling Car Population Development

The development of the car population for different scenarios is calculated using an iterative deregistration and production process. In the first step, the vehicle population (starting with the 2019 population from [22]) for each year of manufacture is reduced according to the deregistration rate for the respective age. Subsequently, the age groups are shifted back one year, and the vehicle production of the considered year is set to maintain a constant total vehicle population. The vehicle deregistration

rate is scaled from the real data shown in Figure 5 in order to match the prognosis of newly registered cars between 2020 and 2030 and of the "proKlima" [20] scenario.

In the next step, the modified scenarios (shown in Figure 6) are derived from the baseline by adapting the deregistrations only. First, the deregistration rate is decreased to reflect the reduced registrations due to owners' financial uncertainty during the COVID-19 pandemic. The total vehicle registrations in 2020 are decreased by 23% compared to 2019 [1]. To simulate a subsidy in 2021, the deregistration of vehicles older than 10 years is increased to match the total number of subsidized vehicles. For the baseline scenario, this number is set to the predicted new registration decrease due to the COVID-19 pandemic (0.72 million). In Section 4.4, we discuss the impact of the number of subsidized cars; therefore, additional analyses of twice and four times the baseline amount (1.43 and 2.87 million) were made.

This approach considers the reduced vehicle production in subsequent years after the subsidy is introduced (pull-out effect) and the increased total vehicle production due to early deregisterations compared to a scenario without a subsidy.

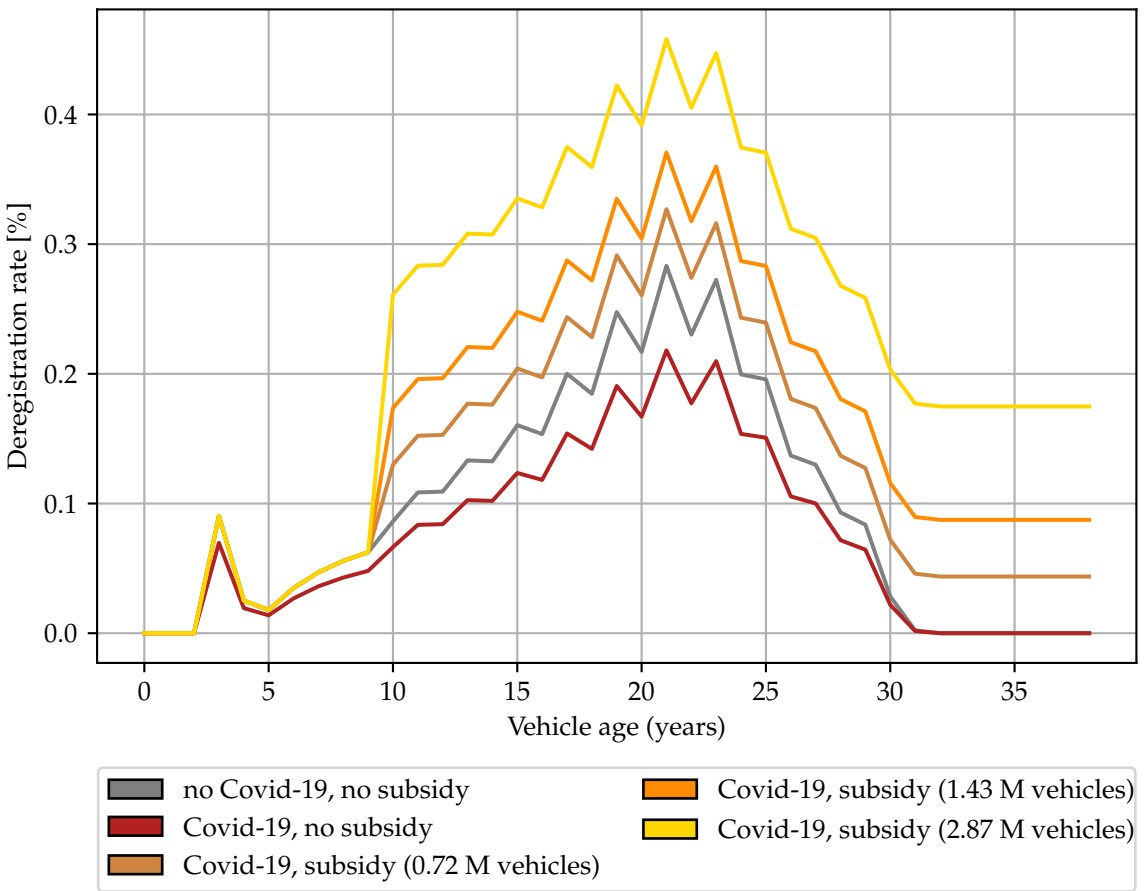

**Figure 6.** Deregistration rate distributions of different scenarios.

Figure 7 shows the annual and the accumulated vehicle registrations for different scenarios. The economic uncertainty due to the COVID-19 pandemic reduces the total number of vehicle registrations by forcing owners to keep their vehicles longer. On the contrary, the subsidy leads to earlier replacements and, therefore, increases the new registrations. As already explained for Figure 5, there is a first peak of vehicle decommissioning after three years.

This can also be seen in the new registrations, as the additional vehicles produced due to the subsidy also lead to an increased number of vehicles being taken out of service three years after the subsidy. In turn, this leads to an increase in new purchases that can be seen four years after the subsidy.



This time delay is due to the fact that we model the transition from one year to the next, and thus, the ages of the vehicles in the previous year are taken into account for the scrapping and new purchases in the year under consideration.

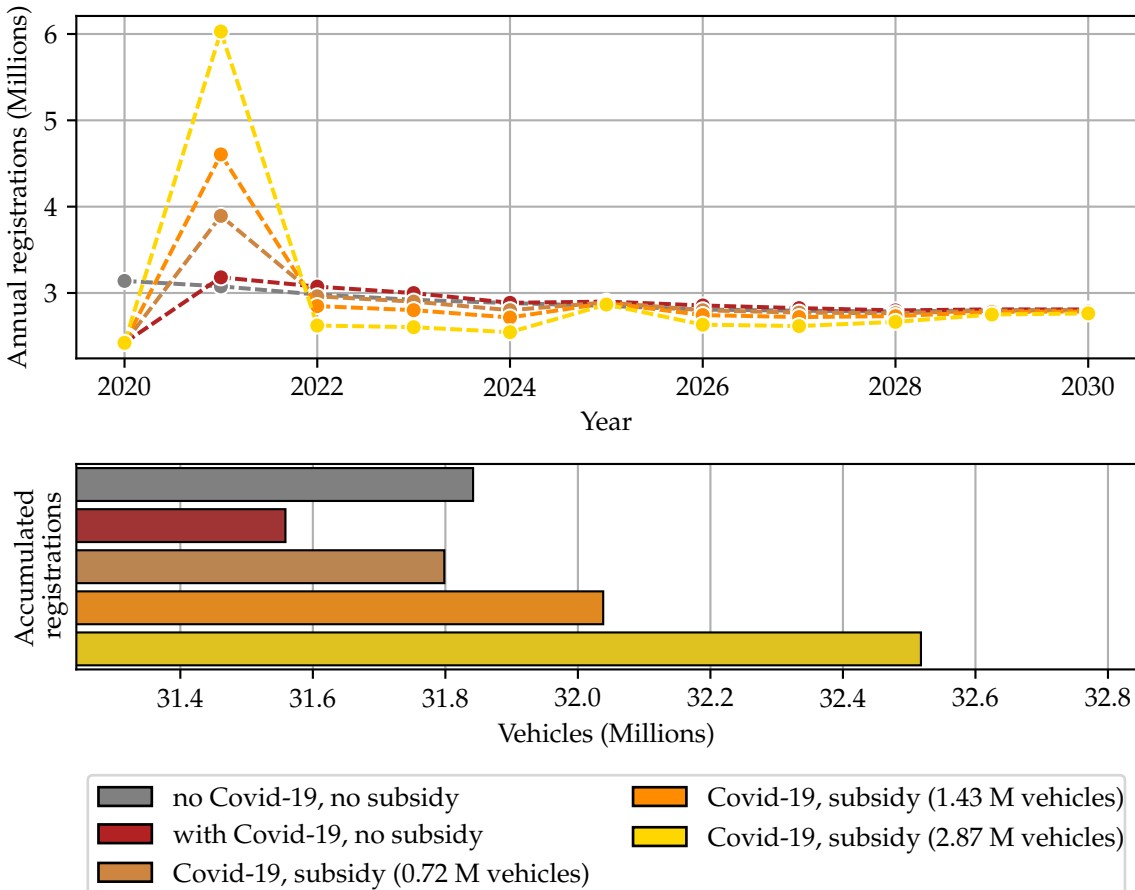

**Figure 7.** Annual and accumulated vehicle registrations.

### 3.5. Life-Cycle Emissions

To perform a cradle-to-grave life-cycle analysis, the production, use, and EoL phases are considered. The focus is primarily on the calculation of greenhouse gas emissions ($CO_2$ equivalent emissions, 20-year global warming potential (GWP 20)). In this analysis, the direct $CO_2$ emissions of the combustion engines in the use phase are set identically to their $CO_2$ equivalent emissions.

The amounts of production and EoL emissions are assumed to be constant for the upcoming years. They are only dependent on the segment and power train type of the vehicle and do not change with the production and disposal date.

In order to analyze the life-cycle emissions of the drive train technologies per segment, specific input data (weight, direct $CO_2$ emissions, fuel, and electric energy consumption) are needed. Complete data for gasoline and diesel vehicles are available. For PHEVs and BEVs, the required input is derived from defined reference vehicles. Currently, in some segments, there are no vehicles available. In these cases, input data are interpolated from adjacent segments.

#### 3.5.1. Production

All production analyses were performed with the Ecoinvent 3.5 database [23]. The cutoff allocation was used and all location settings were set to global.

Petrol and Diesel

To perform a life-cycle analysis for gasoline and diesel vehicles, the curb weight was used as the input value (Table 2).

**Table 2.** Weight distribution of vehicles with combustion engines [24].

| Segment | Curb Weight [kg] |
|---|---|
| mini class | 1038 |
| small class | 1191 |
| compact class | 1389 |
| middle class | 1617 |
| upper middle class | 1831 |
| upper class | 2035 |
| SUVs | 1506 |
| mini class—vans | 1514 |
| large vans | 1754 |

It was assumed that there is no curb weight difference between gasoline and diesel vehicles. The emissions were calculated with the processes *passenger car production, gasoline* and *passenger car production, diesel* in Ecoinvent. To consider production only, the Ecoinvent data (flow) *manual dismantling of a used passenger car with internal combustion engine* were excluded.

Battery Electric Vehicles (BEVs)

In order to perform the production analysis for the BEVs, the curb weight is divided into the battery weight and the remaining vehicle weight. The battery weight is calculated based on the battery capacity. Therefore, a constant energy density for all BEVs is defined.

A representative energy density at a battery packaging level is defined by averaging the values of the VW E-golf (113 Wh/kg) [25] and the 2017 Tesla Model S P100D (160 Wh/kg) [26]. This results in an energy density of 136 Wh/kg, which is used to calculate the battery weight of all BEVs based on the battery capacity. For each segment, a reference BEV is defined based on the number of newly registered vehicles in 2019 [27] (Table 3). In the SUV segment, the Hyundai Kona Electro and the Audi E-Tron both had nearly the same registration numbers in 2019. Since the share of compact SUVs is expected to continue growing [27], the Hyundai Kona Electro is chosen as the reference model.

**Table 3.** Weight distribution of battery electric vehicles (BEVs).

| Segment | Curb Weight (kg) | Battery Capacity (kWh) | Battery Weight (kg) | Remaining Curb Weight (kg) | Reference |
|---|---|---|---|---|---|
| mini class | 1095 | 17.6 | 129 | 966 | smart EQ fortwo. [28] |
| small class | 1345 | 42 | 308 | 1037 | BMW i3 [29] |
| compact class | 1545 | 40 | 293 | 1252 | Nissan Leaf [30] |
| middle class | 1611 | 55 | 404 | 1207 | Tesla Model 3 [31] |
| upper middle class | - | - | - | - | - |
| upper class | 2290 | 100 | 734 | 1556 | Tesla Model S [31] |
| SUVs | 1593 | 39 | 288 | 1305 | Hyundai Kona Elektro [32] |
| mini vans | 1610 | 39 | 288 | 1322 | Kia e-soul [33] |
| large vans | - | - | - | - | - |

The emissions emitted from the BEV battery pack production were determined with the process *battery production, Li-ion, rechargeable, prismatic*. This process also considers the transportation of the single cells from Peking to Amsterdam by ship and a 1000 km transportation route within Europe by truck.

For the remaining electric vehicles, the process *passenger car production, electric, without battery* [34] was used with the remaining curb weight from Table 3.

Plug-in Hybrid Electric Vehicles (PHEVs)

In this analysis, the PHEV is a combination of a conventional and an electric vehicle. Therefore, the PHEV consists of a battery, electric power train components, and the remaining vehicle, which contains all components of the combustion power train. This analysis is based on manufacturers' data from reference vehicles (Table 4).

**Table 4.** Manufacturers' data for plug-in hybrid electric vehicles (PHEVs).

| Segment | Curb Weight (kg) | Battery Capacity (kWh) | Maximum Power of Electric Drive Train (kW) | Reference |
|---|---|---|---|---|
| mini class | - | - | - | - |
| small class | 1660 | 7.6 | 65 | Mini Cooper SE Countryman [35] |
| compact class | 1750 | 8.8 | 65 | BMW 225xe Active Tourer [36] |
| middle class | 1780 | 9.8 | 50 | Kia Optima Plug-In Hybrid [37] |
| upper middle class | 1910 | 11.2 | 62 | BMW 530 e [38] |
| upper class | 2170 | 14.1 | 100 | Porsche Panamera 4 E-Hybrid [39] |
| SUVs | 1971 | 13.8 | 70 | Mitsubishi Outlander [40] |
| mini vans | 1725 | 15.6 | 75 | Mercedes B 250 e [41] |
| large vans | - | - | - | - |

Similarly to the BEV batteries, the PHEV battery weight is calculated based on the battery capacity with a constant energy density for all types of plug-in vehicles. We consider all batteries at the packaging level. Since the PHEV batteries are significantly smaller than the BEV ones, an assumption of the same energy density is not suitable.

Therefore, the mean value of the energy densities of the Kia Optima (75 Wh/kg) [37] and the Mercedes B 250 e (104 Wh/kg) [42] of 89.5 Wh/kg are assumed for all PHEVs.

In addition, the electric power train components, such as an electric motor, a charger, and cables, are considered. Due to lacking data on the actual weight of these components, the weight is scaled with the maximum power of the electric power train. In this analysis, the baseline is made according to the default value in Ecoinvent of a 100 kW power train with a weight of 70 kg. The resulting parameters (Table 5) are used in Ecoinvent for the production analysis.

**Table 5.** Calculated weight distribution of PHEVs.

| Segment | Remaining Curb Weight (kg) | Battery Weight (kg) | Electric Power Train Weight (kg) | Reference |
|---|---|---|---|---|
| mini class | - | - | - | - |
| small class | 1525 | 85 | 50 | Mini Cooper SE Countryman [35] |
| compact class | 1601 | 99 | 50 | BMW 225xe Active Tourer [36] |
| middle class | 1632 | 110 | 39 | Kia Optima Plug-In Hybrid [37] |
| upper middle class | 1724 | 125 | 50 | BMW 530 e [38] |
| upper class | 1935 | 158 | 77 | Porsche Panamera 4 E-Hybrid [39] |
| SUVs | 1763 | 154 | 54 | Mitsubishi Outlander [40] |
| mini vans | 1393 | 174 | 58 | Mercedes B 250 e [41] |
| large vans | - | - | - | - |

For the battery pack production of the PHEVs, the same method as that used for the BEVs was applied. The process *electric motor production, vehicle* was used to determine the emitted emissions to produce the components of the electric power train.

The remaining part of the vehicle is assumed to be a gasoline engine car. Therefore, the process *passenger car production, petrol* was applied to analyze the production.

### 3.5.2. Use Phase

The emissions emitted during the use phase are composed of the emissions based on fuel and electric energy consumption, the well-to-tank emissions, and the emissions due to maintenance of the vehicle and the road infrastructure.

In 2019, the average annual mileage of passenger cars was 13,602 km [43]. In this model, it is assumed that, in the future, every car will have the 2019 average annual mileage.

### Combustion Engine Vehicles

The use phase emissions of the gasoline- and diesel-powered vehicles are dependent on the vehicle's segment and age. Based on these criteria, the KBA [44] provides a database of the average direct $CO_2$ emissions and fuel consumption, which are displayed in the supplementary materials in Table S1. In this work, they are assumed to be identical to the $CO_2$ equivalent emissions.

### Battery Electric Vehicles (BEVs)

In this analysis, the BEVs cause indirect $CO_2$ emissions because of their consumed electric energy during the use phase.

For each segment, a reference BEV is defined based on the number of newly registered vehicles in 2019 [27]. If a model in this segment would have a dominant share of the newly registered cars, it is considered as the reference BEV. Otherwise, the two models with the highest number of newly registered vehicles in the segment are selected, and a weighted average of the energy consumption (weight factor) according to their new registration numbers is built (Table 6).

**Table 6.** Energy consumption of BEVs.

| Segment | Reference Model | Energy Consumption (kWh/100 km) | Weight Factor | Weighted Average Energy Consumption (kWh/100 km) |
|---|---|---|---|---|
| mini class | smart EQ fortwo. [28] | 14.0 | 1 | 14.0 |
| small class | BMW i3 [29] | 13.1 | 0.66 | 14.5 |
| | Renault Zoe [45] | 17.2 | 0.34 | |
| compact class | VW E Golf [46] | 12.9 | 0.64 | 14.4 |
| | Nissan Leaf [30] | 17.1 | 0.36 | |
| middle class | Tesla Model 3 [31] | 16.0 | 1 | 16.0 |
| upper middle class | - | - | - | - |
| upper class | Tesla Model S [31] | 19.0 | 1 | 19.0 |
| SUVs | Hyundai Kona Electro [32] | 15.0 | 0.51 | 18.6 |
| | Audi E-Tron [47] | 22.4 | 0.49 | |
| mini vans | Kia e-soul [33] | 15.6 | 1 | 15.6 |
| large vans | - | - | - | - |

In 2019, there were not any BEVs newly registered in the upper middle class or large van segments. Since the market for electric vehicles will grow [20], we assume that future electric vehicles will also be available in these segments. Based on the curb weight of the combustion engine vehicles, we estimate that upper middle class and large vans have the same energy consumption. The electric energy consumption of upper middle class is assumed as an average of those of the middle class and upper class.

Plug-In Hybrid Electric Vehicles (PHEVs)

To determine comparable $CO_2$ emissions from plug-in hybrids, a combination of the direct exhaust emissions and the energy consumption of the electric drive train is made (Table 7).

The manufacturers' data for the reference vehicles either rely completely on the New European Driving Cycle (NEDC) or were measured with the Worldwide Harmonized Light Vehicles Test Procedure (WLTP) and were transferred back to the NEDC. The NEDC considers half of the driving cycle using the combustion engine and the other half using the electric power train. In these data, the electric energy consumption is considered emission-free. In order to correct this effect, we add the appropriate indirect emissions for electric power consumption.

**Table 7.** $CO_2$ emissions and energy consumption of PHEVs.

| Segment | $CO_2$ Emissions (Combined) (g/km) | Fuel Consumption (L/100 km) | Energy Consumption (kWh/100 km) | Reference Model |
|---|---|---|---|---|
| mini class | - | - | - | - |
| small class | 45 | 2,0 | 14.0 | Mini Cooper SE Countryman |
| compact class | 42 | 1.9 | 13.5 | BMW 225xe Active Tourer |
| middle class | 37 | 1.6 | 12.2 | Kia Optima Plug-In Hybrid |
| upper middle class | 42 | 1.8 | 14.8 | BMW 530 e |
| upper class | 62 | 2.7 | 16.1 | Porsche Panamera 4 E-Hybrid |
| SUVs | 40 | 1.8 | 14.8 | Mitsubishi Outlander |
| mini vans | 32 | 1.4 | 14.7 | Mercedes B 250 e |
| large vans | - | - | - | - |

Indirect Emissions

In this model, the consumed electric energy causes indirect emissions based on the specific carbon dioxide emissions of the German electricity mix. In 2020, the generation of one kWh electric energy is predicted to cause 432 g $CO_2$ eq. emissions [48].

Pehnt et al. [48] calculated 318 g $CO_2$ eq. emissions for the electricity mix in 2030; the value here decreases linearly for the years 2020 to 2030. The well-to-tank factors are used to determine the upstream emissions of consuming fuel. Here, Schallaboeck et al. [49] stated that 685 g or 408 g $CO_2$ equivalent emissions produce one liter of gasoline or diesel, respectively. The data for the fuel consumption are displayed in the supplementary materials in Table S2. All PHEVs are assumed to have a gasoline engine.

Maintenance

During the use phase, the replacement of spare vehicle parts and the maintenance of the road network are considered. To calculate the vehicle maintenance, the flow *passenger car maintenance* was used in Ecoinvent and was adapted according to the vehicle's weight and power train type [50]. We followed the same approach as in the Agora (2019b) [51] "basic scenario", assumed for each vehicle a lifespan of 150,000 km, and did not consider any battery exchange for BEVs and PHEVs.

In order to determine the maintenance of the road network, the flow *road maintenance* is used. This introduces a weight-dependent contribution during the use phase.

3.5.3. End-of-Life

In this model, it is assumed that every deregistered vehicle is eventually disposed of. The equivalent $CO_2$ emissions due to this process were determined in Ecoinvent using the flows *manual dismantling of used passenger car with internal combustion engine, treatment of used glider, passenger car, shredding, treatment of used internal combustion engine, shredding, treatment of used powertrain for electric passenger car, dismantling,* and *market for used Li-ion battery* according to [34].

### 3.6. Subsidy Concepts

Three subsidy concepts are introduced in this chapter. They reflect aspects of the current discussions about the possible subsidy scope. Höpfner et al. [3] showed that 84% of the subsidy from the "environmental bonus" in 2009 was used to purchase small cars (mini class, small class, and compact class). Therefore, 84% of all new registrations due to the subsidy will happen in the mini class, small class, and compact class categories. For this distribution, the predicted new car registration shares for 2021 are used. All segment shares are evenly scaled in order to reach an 84% share in the small cars category. The impact of this effect is discussed in Section 4.4.1.

The distribution shown in Figure 8 is used as a baseline for all subsidy concepts. In Section 4.4.1, the influence of this distribution is discussed.

Similarly to the "environmental bonus" in 2009, a mandatory requirement to earn the subsidy is a minimum vehicle age (in this case, 10 years).

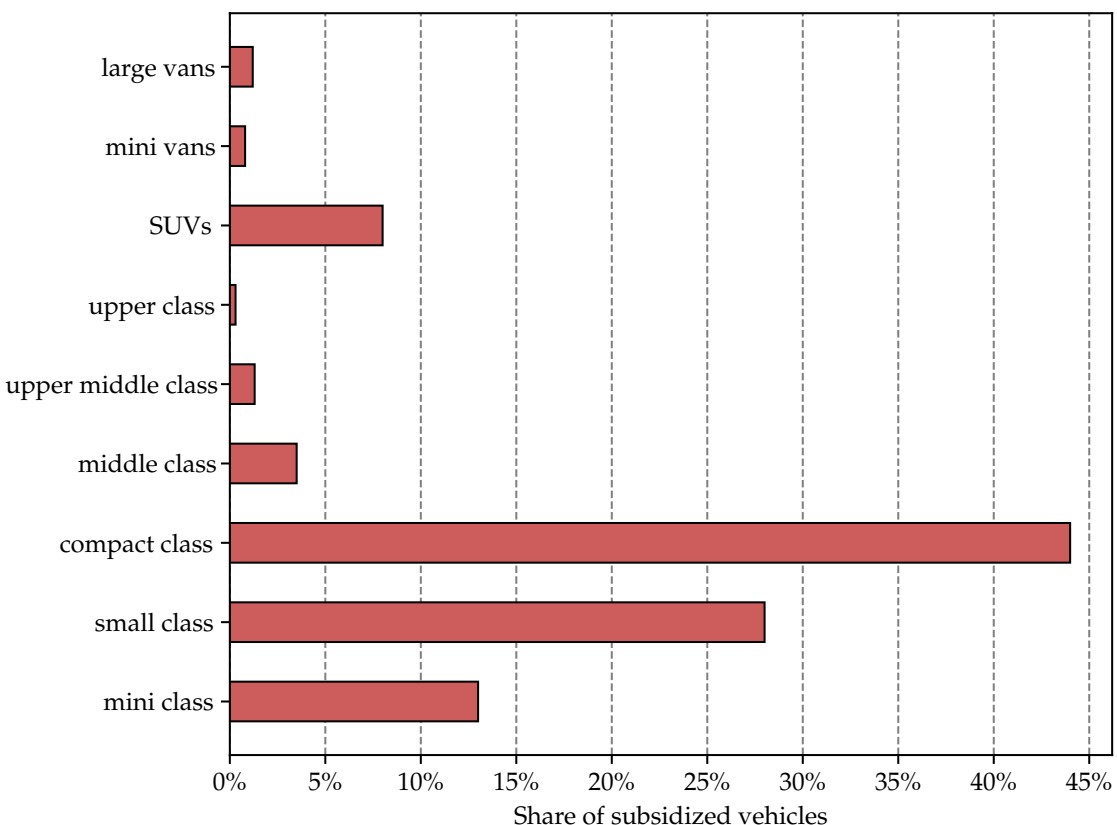

**Figure 8.** Subsidized vehicle distribution of the "broad funding" concept.

### 3.6.1. "Broad Funding" Concept

In this concept, the subsidy is not restricted to a certain vehicle type. Except for the higher demand for the mini, small, and compact class segments, the same composition of newly registered cars as predicted without the subsidy will occur. The subsidy only increases the total number of newly registered and deregistered cars. For every newly registered car due to the subsidy, a car in the same segment, manufactured in 2010 or earlier, is disposed of. Vehicle purchases in higher segments are not subsidized.

### 3.6.2. "Innovation" Concept

With this concept, only BEVs are subsidized. For the baseline model, the same distribution of subsidized cars as in Section 3.6 is considered, except only BEVs will be newly registered due to the subsidy.

### 3.6.3. "Downsizing" Concept

Lastly, a concept is shown that only allows a subsidy if the new car is at least one segment smaller (defined by the segment's $CO_2$ emissions) than the one traded in. The respective segment shifts in the "downsizing" concept are displayed in Table 8.

**Table 8.** Segment shifts in "downsizing" concept.

| Segment of Vehicle Traded in | Segment of New Vehicle |
|---|---|
| mini class | - |
| small class | mini class |
| compact class | small class |
| middle class | compact class |
| upper middle class | middle class |
| upper class | upper middle class |
| SUVs | middle class |
| mini vans | middle class |
| large vans | mini vans |

As per the $CO_2$ emission-based downsizing definition, the SUVs and mini vans are shifted to the middle class segment.

## 4. Results

The goal of this work is to analyze the impact of a subsidy on the $CO_2$ eq. emissions of the German passenger car fleet. This is achieved by considering categorized vehicle emissions and fleet composition data.

### 4.1. Life-Cycle Analysis

The life cycle of a vehicle consists of production, maintenance, use phase, and EoL processes. For each element, the respective emissions are calculated per segment and per drive train technology.

### 4.1.1. Production

Using the process described in Section 3.5.1, an analysis of the production emissions for each vehicle is made. The results for each vehicle class and drive train technology are shown in Figure 9.

The production of BEVs and PHEVs results in higher $CO_2$ equivalent emissions than the production of conventional vehicles. This phenomenon emerges in every segment and can be attributed, in addition to the higher vehicle weight of BEVs and PHEVs, to the greater effort of producing batteries and electrical parts. The BEV production effort increases substantially for larger segments as more battery capacity is implemented. This effect is less pronounced for PHEVs.

### 4.1.2. Maintenance

The emissions from the vehicle and road maintenance per driven kilometer (Figure 10) scale mainly with the vehicle weight.

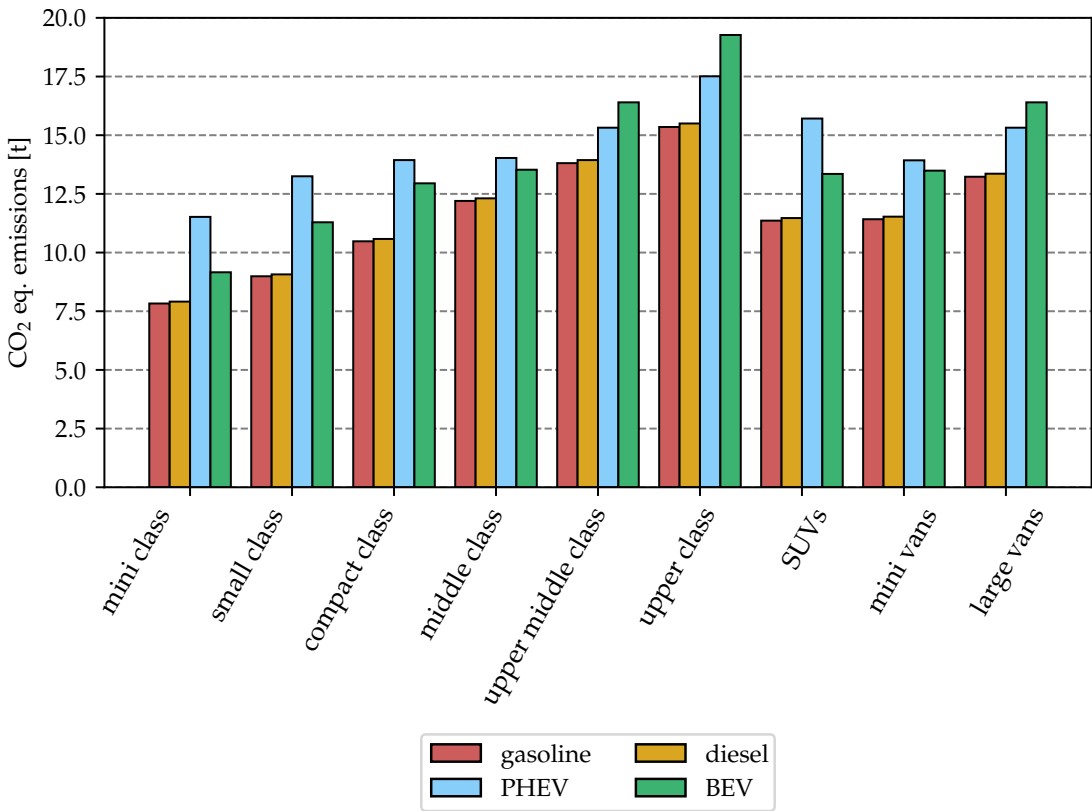

**Figure 9.** Production emissions per vehicle, segmented by class and drive train technology.

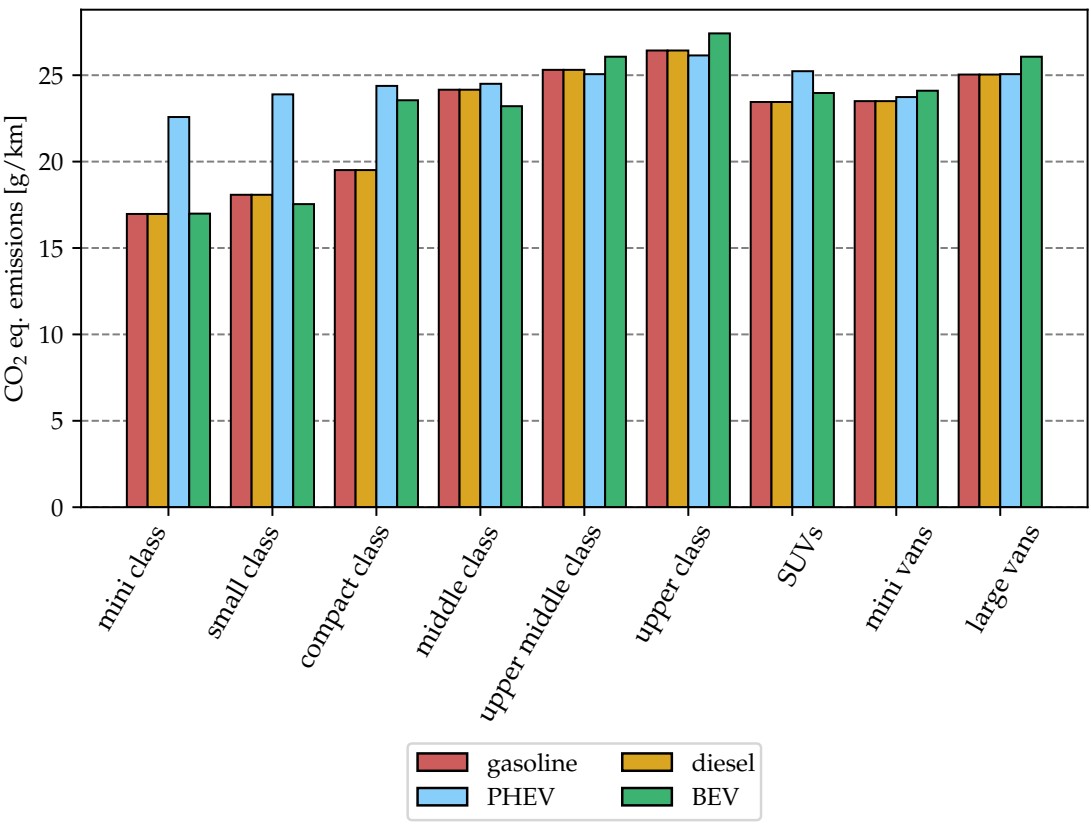

**Figure 10.** Maintenance emissions.

### 4.1.3. Use Phase

The $CO_2$ equivalent emissions depend on the vehicle specifications. In addition to the direct exhaust emissions, vehicles with combustion engines cause indirect emissions via the procurement of the fuel. Figure 11 shows the combined specific $CO_2$ eq. emissions per kilometer in 2019.

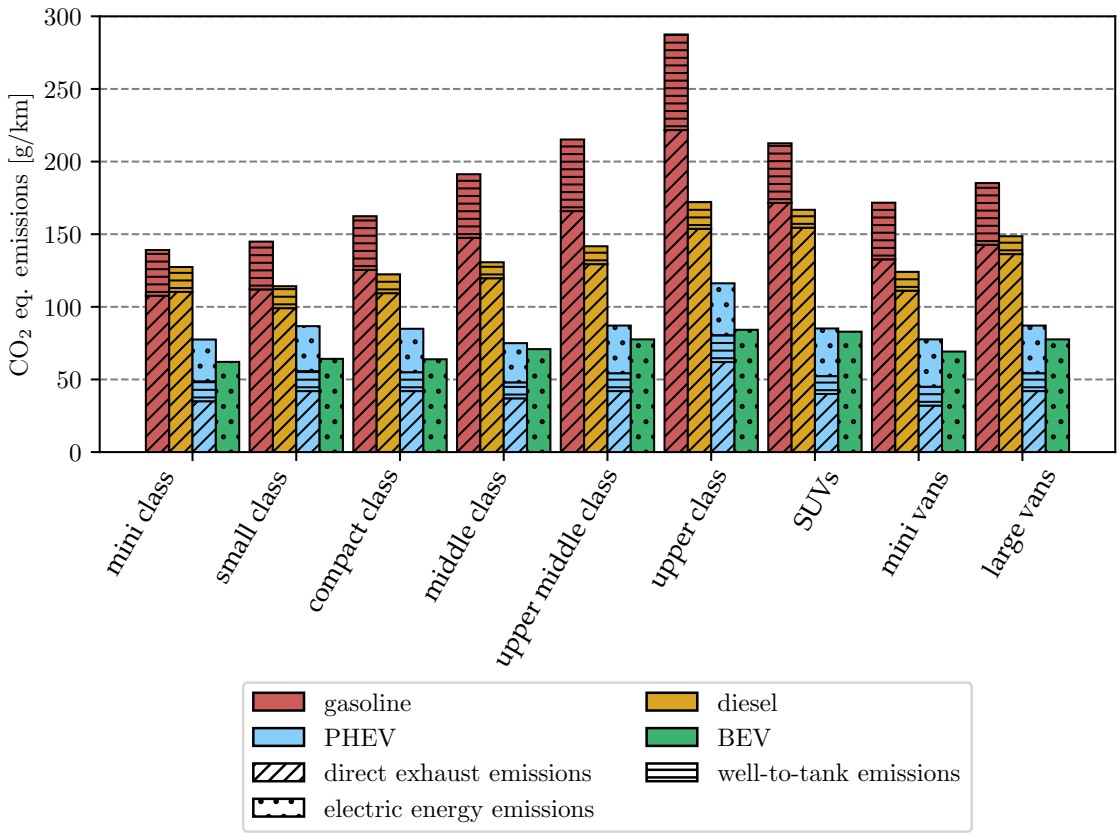

**Figure 11.** Composition of $CO_2$ eq. emissions per driven kilometer.

Vehicles with a gasoline engine have the highest specific emissions across every segment, followed by diesel engine vehicles. The BEVs have the lowest specific emissions of the considered drive train technologies in all segments.

### 4.1.4. End of Life

Due to the high recycling effort for the batteries, we see increased EoL emissions for the BEVs and PHEVs compared to the conventionally powered vehicles (Figure 12). The higher battery capacity in larger segments leads to higher EoL emissions for the BEVs compared to the PHEVs. Again, the vehicle weight is an important driver of EoL emissions.

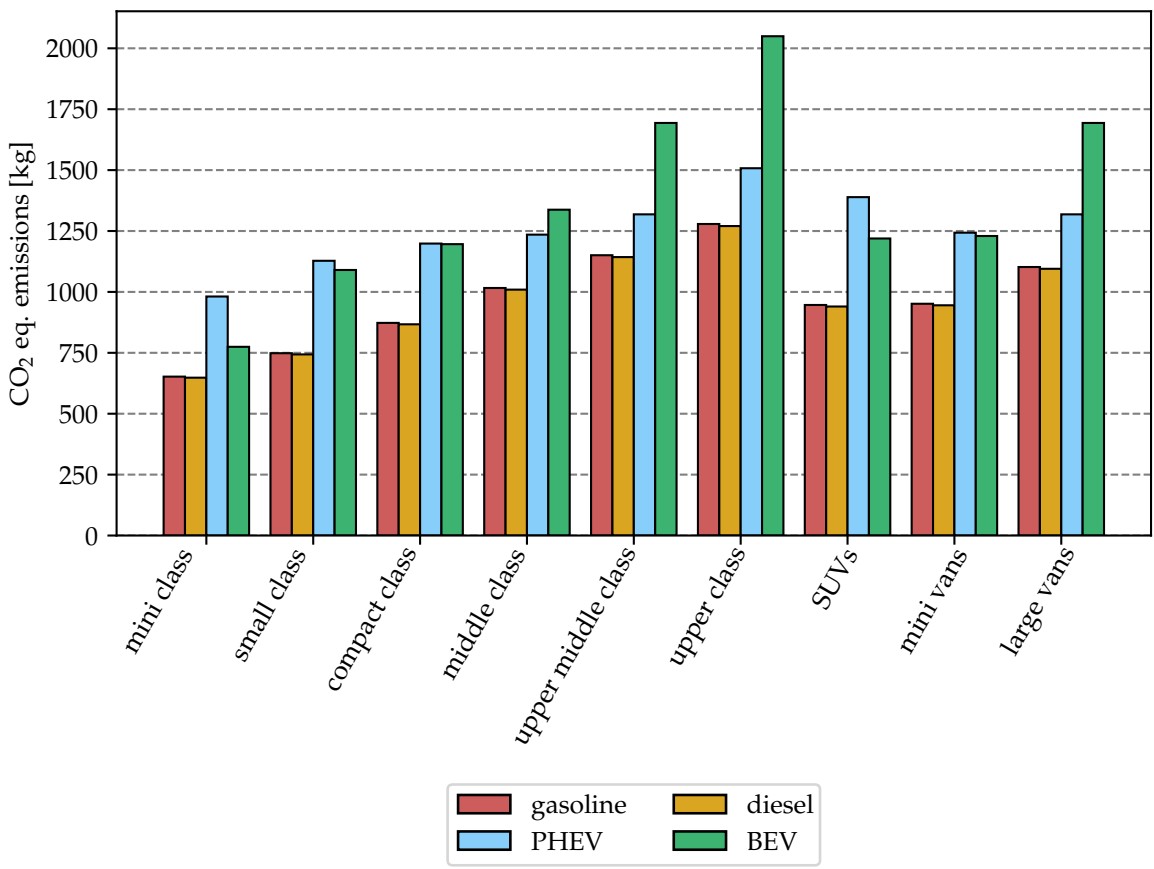

**Figure 12.** End-of-life emissions per vehicle, segmented by class and drive train technology.

### 4.2. Subsidy's Impact on Fleet Emissions for 2019–2030

Using a combination of the results from Section 4.1 and the German car population scenario from Section 3.3, a prediction of the $CO_2$ equivalent emissions emitted from the German passenger car system until 2030 is made. The emissions decrease in 2020 due to reduced vehicle production (Figure 13). As shown in Figure 7, the vehicle production increases in the subsequent years due to the subsidy. Therefore, higher emissions are expected before the vehicle fleet benefits from the increased efficiency that leads to reduced annual emissions.

The cumulative $CO_2$ eq. emissions of the baseline scenario without a subsidy (Figure 13) are compared to the cumulative emissions of the various subsidy concepts. Each concept is calculated with 0.72 million subsidized vehicles. The developed model enables the analysis of any number of subsidized vehicles.

Figure 14 shows the comparison of the different subsidy concepts relative to the baseline scenario. As a result, the accumulated differences of the $CO_2$ equivalent emissions are shown for the respective scenarios.

The cumulative difference shows significant additional emissions in 2021 for all subsidy concepts. The effect displays the higher number of new vehicles and the related production emissions for this year. The emission savings are dependent on the subsidy concept.

The "innovation" subsidy clearly lowers the $CO_2$ equivalent emissions of the German passenger car transport system until 2030 the most out of all considered concepts. Even with the highest $CO_2$ equivalent emissions in 2021, the "innovation" concept reaches the break-even earliest in the year 2025. The additional $CO_2$ equivalent emissions of the "innovation" concept in 2021 are 8.6% higher than the "broad funding" and 20.8% higher than the "downsizing" concepts. This effect can be attributed to the higher production effort for electric vehicles, as shown in Figure 9.

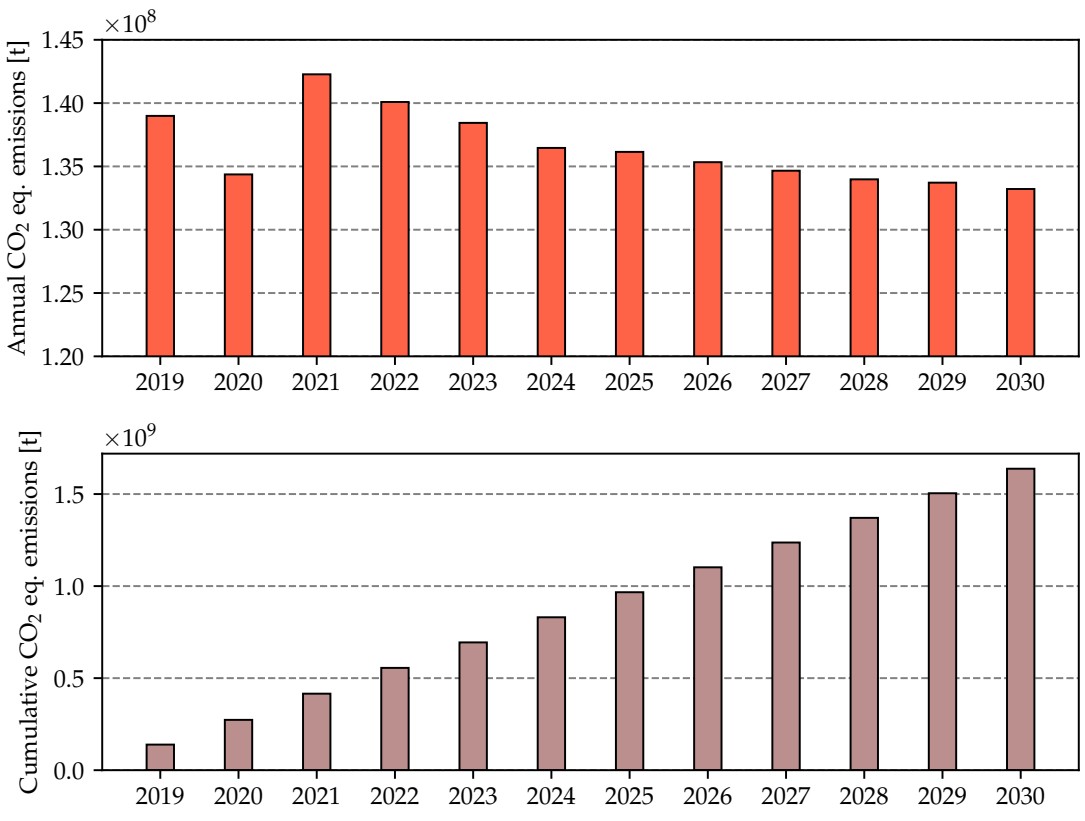

**Figure 13.** Life-cycle $CO_2$ eq. emissions of the baseline scenario without a subsidy.

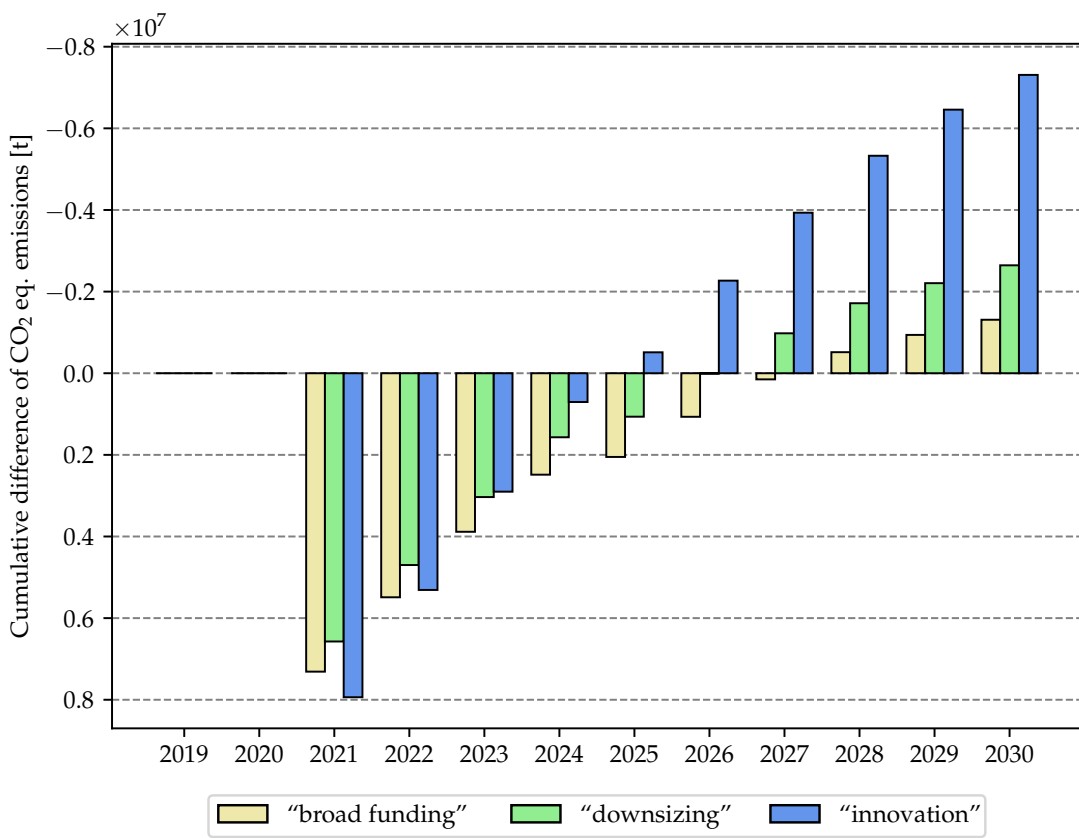

**Figure 14.** Cumulative difference of $CO_2$ eq. emissions compared to the scenario without a subsidy.

Despite having the highest $CO_2$ equivalent emissions in 2021, the accumulated $CO_2$ equivalent emission savings until 2030 of the "innovation" concept are 285.7% higher than the "downsizing" and 576.7% higher than the "broad funding" concepts. Compared with the scenario without a subsidy, the "innovation" concept reaches a cumulative difference of $-7.56 \times 10^6$ t $CO_2$ equivalent emissions until 2030.

In the "downsizing" concept, compared to the "broad funding" concept, smaller vehicles are produced. Therefore, in 2021, the $CO_2$ equivalent emissions of the "downsizing" concept are lower than the emissions in the "broad funding" concept. In addition, the "downsizing" concept reaches the break-even point in 2027, one year earlier than the "broad funding" subsidy concept.

### 4.3. Sensitivities

Due to the complexity of the model, dependencies on key parameters are reviewed and their respective impacts on the subsidy concepts are determined.

### 4.4. Number of Subsidized Cars

As shown in Section 3.4, we assume that the number of subsidized cars matches the predicted registration decrease in 2020 due to the COVID-19 pandemic. To determine the impact of the number of subsidized cars (~0.7 million), this value is doubled (~1.4 million) and quadrupled (~2.8 million).

As expected, the cumulative difference of $CO_2$ eq. emissions in 2030 is proportional to the number of subsidized cars, which can be seen in Figure 15.

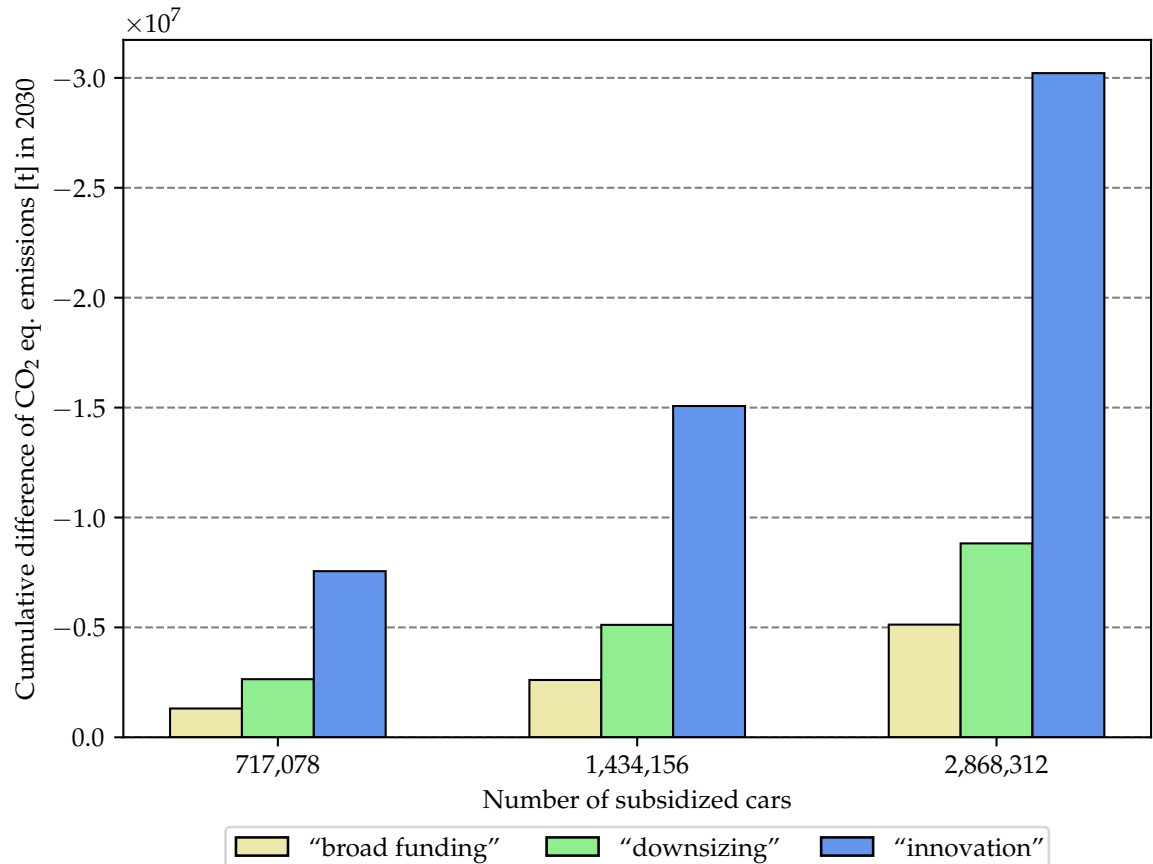

**Figure 15.** Number of subsidized cars.

### 4.4.1. Vehicle Distribution

As discussed in Section 3.6, the "environmental bonus" in 2009 was primarily used to purchase smaller vehicles. This can be attributed to the fairly high share of the subsidy compared to the purchase price.

It would be conceivable to design the subsidy in a way that the amount is determined by a fixed percentage of the purchase price. It is assumed that this leads to a vehicle distribution of the subsidized cars proportional to the predicted new passenger car registrations in 2021.

Figure 16 shows the difference between a scenario proportional to 2021 and a scenario shifted to smaller cars. The modified distribution of subsidized vehicles has an impact on the results.

As shown in Table 9, all subsidy concepts are less efficient when using the distribution proportional to 2021. In this context, the "broad funding" concept even increases the $CO_2$ eq. emissions compared to the no-subsidy baseline. As a result, subsidizing smaller cars significantly increases the efficiency of the subsidy.

**Table 9.** Difference of cumulative $CO_2$ eq. emissions compared to the scenario without a subsidy in 2030.

| Subsidy Concept | Difference to "Without Subsidy" Baseline | |
| --- | --- | --- |
| | Shifted Distribution to Smaller Cars | Proportional to 2021 Distribution |
| "broad funding" | −0.080% | 0.037% |
| "downsizing" | −0.161% | −0.017% |
| "innovation" | −0.462% | −0.401% |

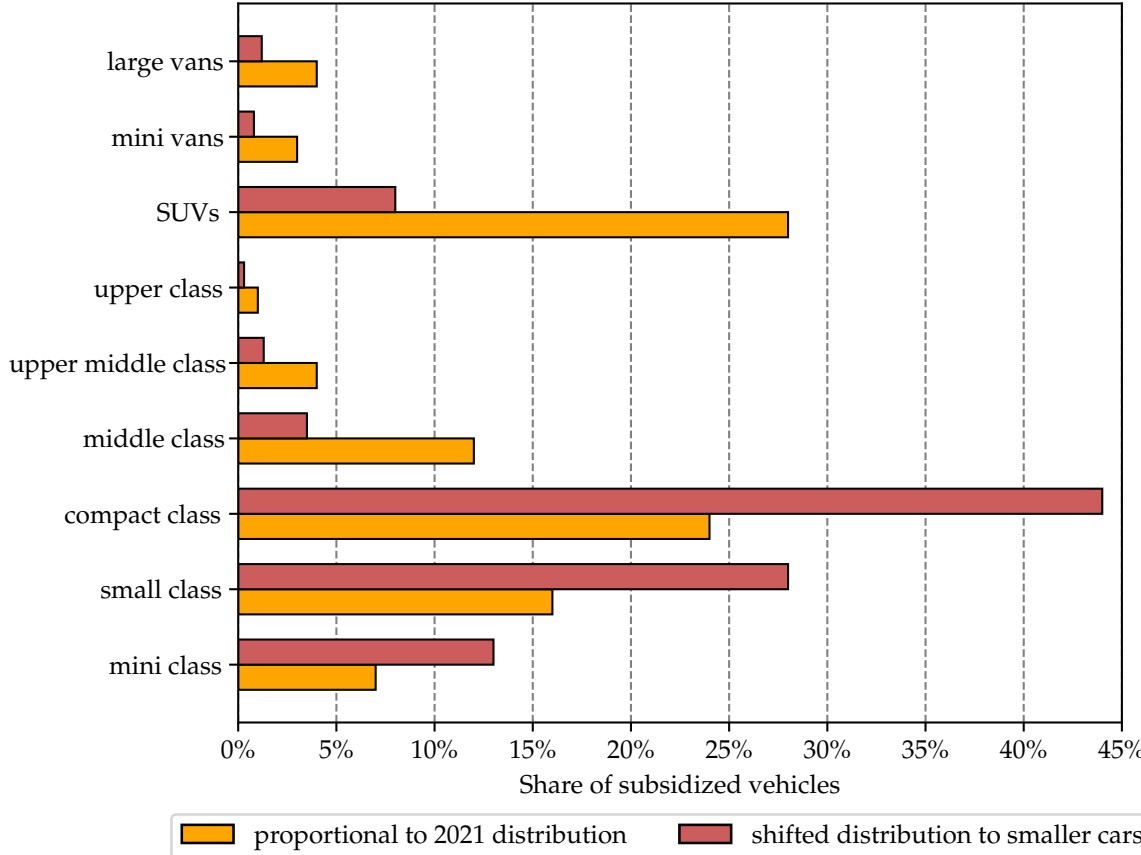

**Figure 16.** Subsidized car distribution for two scenarios under investigation.

### 4.4.2. New Vehicle Registrations

Drive Train Technologies

To predict the German car population until 2030, we rely on Section 3.3 and on the "ProKlima" scenario stated in Agora (2018) [20] to describe the trend of the drive train technologies. To estimate the impact of the subsidy for a baseline scenario with a higher share of newly registered PHEVs and BEVs, an analysis with the "ProKlima Plus" scenario is made.

In Table 10, the values for the drive train technology development of the different scenarios can be seen. To determine the share of newly registered cars for each year, a linear interpolation between the displayed values is made.

**Table 10.** Annual share of newly registered cars.

| Drive Train Technology | "ProKlima" | | "ProKlima Plus" | |
|---|---|---|---|---|
| | 2025 | 2030 | 2025 | 2030 |
| gasoline | 57% | 42% | 27% | 28% |
| diesel | 29% | 19% | 28% | 10% |
| PHEV | 8% | 22% | 20% | 29% |
| BEV | 6% | 18% | 25% | 47% |

Figure 17 shows the cumulative $CO_2$ eq. emission difference of the "innovation" concept with the two different assumptions for the annual newly registered cars until 2030. It is shown that the subsidy loses efficiency if the baseline share of PHEVs and BEVs rises. The cumulative $CO_2$ eq. emission difference will be smaller due to the overall more efficient German passenger car fleet with the "proKlima Plus" scenario.

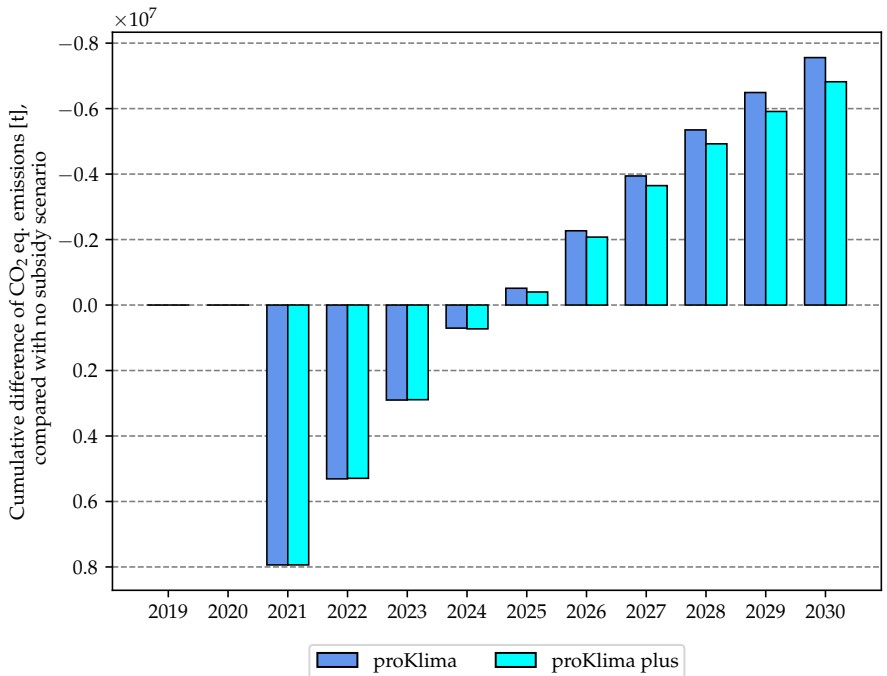

**Figure 17.** Cumulative $CO_2$ eq. emission difference between the "no subsidy" baseline and the "innovation" concept using the "proKlima" and "proKlima Plus" registration predictions.

Segments

In Section 3.3, an assumption of the development of the segment composition of newly registered cars is introduced. Here, we assume that the SUV segment will reach a 40% share of the annual newly

registered cars in 2030. Since this parameter is uncertain, analysis with either a 30% or 50% proportion of SUVs of the newly registered cars in 2030 is made. For the "innovation" concept, the 30% SUV share scenario generates 2.40% less $CO_2$ eq. emissions savings compared to the 40% SUV share baseline due to the overall more efficient car fleet. On the contrary, the cumulative $CO_2$ eq. emissions savings of the "innovation" concept in the 50% SUV share scenario are 0.84% higher compared to the baseline.

This shows that the results of the study are robust against this parameter because the trend and the statement of the results are not changed by a variation of the SUV share.

## 5. Discussion

The absolute $CO_2$ eq. emissions savings are proportional to the number of subsidized cars. For the results shown, we set the subsidized cars in the model to about 0.72 million cars. This is more conservative than the actual replacements of 1.95 million [3] during the 2009 "environmental bonus".

Our model is able to deal with a variable number of cars, as shown in the sensitivity analysis in Section 4.4. The actual number of subsidized cars would depend on micro- and macroeconomic factors, as well as technology-specific concerns, with different scenarios needing different funding to reach the same total number of replaced vehicles. Additionally, an investigation of the basic conditions, e.g., charging facilities, is needed to match the implementation of the number of subsidized vehicles. However, since we are not conducting an economic analysis, quantifying these factors reliably is out of the scope of this work. Our model can be easily updated to calculate the effects of arbitrary numbers of subsidized vehicles. In case the 2020 program would have the same impact on car sales, the absolute savings grow accordingly.

In the absence of detailed prognosis data for the technological improvements, we assume no efficiency gains regarding the $CO_2$ eq. emissions. The trend of the combustion engines indicates a saturation for the $CO_2$ efficiency [44]. Assuming that the electric drive train technologies will still gain efficiency [52] and more electricity will be generated from renewable sources, the total savings for the "innovation" concept would be higher.

Since only the direct $CO_2$ emissions for the driving during the use phase of combustion engines are available, the actual $CO_2$ eq. emissions are slightly higher. As this effect is not relevant to BEVs, since $CO_2$ eq. emissions are considered for the electricity production, the "innovation" concept would generate marginally higher savings when considering all emissions from the combustion engines.

This paper shows that all considered subsidy concepts eventually have a positive effect on reducing the $CO_2$ equivalent emissions of the German passenger car fleet. This corresponds with both Höpfner et al. [3] and Lenski et al. [13], who stated that the 2009 car subsidy had a positive effect on the environmental performance of the various car fleets.

Contrary to Klößner et al. [8], the long-term effect on the $CO_2$ emissions is positive, as windfall gains (subsidized purchases that would happen without the subsidy) are not included and BEVs and PHEVs are considered. We only expect additional $CO_2$ emissions to occur immediately after the subsidy is introduced due to the increased production, which is overcompensated by savings during the use phase.

To calculate the LCA emissions, we clustered the vehicle fleet into segments to reflect the actual distribution. Agora (2019) [53] used only one reference vehicle for each drive train technology. For comparable segments, the production emissions of the BEVs show similar results. In this work, the gasoline and diesel vehicles result in higher production emissions compared to those in Agora (2019) [53]. This can be attributed to a deviation in parameter settings.

## 6. Conclusions

In order to calculate the cumulative $CO_2$ eq. emissions of the German passenger car fleet from 2019–2030, a granular model was developed. Three different subsidy concepts were introduced and the impact on the $CO_2$ eq. emissions was determined.

Subsidizing the German passenger car system due to the COVID-19 pandemic shows long-term $CO_2$ emissions savings for nearly all investigated scenarios. Only in the sensitivity analysis, the "broad funding" concept leads to slightly increased emissions when using a vehicle distribution proportional to that of 2021. Taking the sensitivity analyses into account, the "innovation" concept shows the most significant emissions savings in the German passenger car system.

Considering the time period 2019–2030 and a total number of 0.72 million subsidized vehicles, the "innovation" concept generates about 7.56 million t less $CO_2$ eq. emissions compared to the scenario without a subsidy. This equates to 0.46% of the total $CO_2$ eq. emissions of the addressed segments in this period. The "downsizing" and "broad funding" concepts create savings of 0.16% and 0.08%, respectively.

In 2021, the increased vehicle production leads to higher $CO_2$ emissions for all subsidy scenarios compared to the scenario without a subsidy. The ecological break-even is reached in 2025 for the "innovation" concept, in 2027 for the "downsizing" concept, and in 2028 for the "broad funding" concept.

If, from an economic and political point of view, a subsidy program for passenger cars in Germany is considered to be desirable, we clearly recommend the exclusive funding of BEVs because the "innovation" concept achieves the highest positive climate impact at the earliest time.

## 7. Outlook

To further investigate the environmental impact of passenger car subsides in Germany, additional greenhouse gases in the use phase and air pollutants must be considered. It seems reasonable to add greenhouse gases with a high GWP. Due to the current discussion on driving bans in German city centers, the subsidy's impact on nitrogen oxides would enhance the model's result.

It is conceivable to examine further subsidy concepts, such as focusing on PHEVs or particular segments.

In the current model, the technological status of the vehicles is assumed to remain at the level of 2019. The indirect emissions of only BEVs and PHEVs will decrease until 2030 due to the slight reduction of the specific carbon dioxide emissions of the German electricity mix. This means that the vehicle weight and direct exhaust emissions of each newly registered vehicle will remain constant in the current model.

In our model, we define a reference vehicle to describe the BEVs and PHEVs due to a lack of data provided by the KBA. To determine the detailed PHEV and BEV model parameters more precisely, the data of all vehicles available need to be consolidated.

As Höpfner et al. [3] notes, the vehicle subsidy in 2009 led to reduced used car sales. This was not implemented in the current model and might affect the disposal rate and the number of newly registered cars.

This paper did not investigate the economic impacts of subsidy scenarios. A detailed market analysis is needed to estimate consumers' buying behavior. To improve the model, it is necessary to further examine the subsidy's impact on the newly registered cars and to address economic effects, like on-top sales and windfall gains.

**Supplementary Materials:** The following are available online at http://www.mdpi.com/2071-1050/12/23/10037/s1: Table S1: Direct $CO_2$ emissions per kilometer, Table S2: Fuel consumption per 100 km.

**Author Contributions:** Conceptualization, L.H. and A.G.; methodology, M.S. and L.H.; software, A.M.S., L.H., and M.S.; validation, A.M.S., A.G., M.S., L.H., and D.G.; investigation, M.S., A.G., A.M.S., and L.H.; resources, A.M.S. and D.G.; data curation, M.S.; writing—original draft preparation, M.S.; writing—review and editing, A.M.S., A.G., L.H., and M.S.; visualization, L.H. and M.S.; supervision, D.G.; funding acquisition, D.G. All authors have read and agreed to the published version of the manuscript.

**Funding:** This research was funded by the Deutsche Forschungsgemeinschaft (DFG, German Research Foundation), grant number 398051144, project title: "Analysis of strategies to fully de-carbonize urban transport".

**Acknowledgments:** We acknowledge the support from the German Research Foundation and the Open Access Publication Fund of TU Berlin.

**Conflicts of Interest:** The authors declare no conflict of interest. The funders had no role in the design of the study; in the collection, analyses, or interpretation of data; in the writing of the manuscript, or in the decision to publish the results.

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
