# Peer review of "Environmental Impact of Subsidy Concepts for Stimulating Car Sales in Germany"

_sustainability, doi:10.3390/su122310037_

Round 1
Reviewer 1 Report
Thank you for the opportunity to read such an interesting manuscript. In my opinion, the article is well structured. All the authors' considerations are well developed and based on well-established scientific literature, data collections and internet resources. The results and the discussion part are clearly presented, and the directions for further research were indicated.
Author Response
Dear Reviewer, Thank you for your valuable feedback. Kind Rgards, Ludger HeideReviewer 2 Report
- This paper studies the environmental impact of three subsidy concepts to stimulate car sales in Germany, which provides a very good reference for policy and decision making.
- The paper is written well. Some concerns need to be addressed before the paper can be published, as listed below.
- The number of subsidies is assumed to be 0.72 million for all the three subsidy concepts. This assumption is not reasonable. It is very likely the numbers of subsidies resulted from the three concepts are different. The “Broad funding” and “Downsizing” concepts are more likely to attract more people to buy new cars. The “Innovation” concept might attract less new sales because of the price and people’s concerns on convenience for charging and other issues. The three concepts are not comparable because of the different number of subsidies. This should be addressed to make the research valid.
- In Line 54, “6.000 km” should be “6 000 km”? Line 279, “13.602 km” should be “13,602 or 13 602 km”
- In Line 134, is “0.2 years” “2 years”? 0.2 years seem too short.
- In Figure 2, what’s the total percent, 100%?
- Figures 9 and 12, is it the emission per vehicle? Should clarify it.
Author Response
Dear Reviewer,
thank you for your valuable feedback. Regarding your points, we have made the following changes and clarifications:
3. (Clarification regarding the number of vehicles)
We believe actually quantifying the amount of vehicles is outside the scope of our work – we answer the question “if n vehicles of the following type are subsidised, what is the environmental impact?”. We have added a paragraph to our discussion section to make this limit more clear.
4. (Decimal Separator Issues)
We have adjusted the manuscript to consistently use commas as thousands separator.
5. (0.2 year difference)
The 0.2 years are the change in average vehicle age between 2017 and 2019. It is derived from two major mobility surveys in the respective years. To make this easier to understand we have clarified the paragraph.
6. (Total percentage)
It is the percentage of vehicles in the scope of this work (82% of overall stock). We have clarified it in the caption.
7. (Figures per vehicle?)
We have clarified it in the caption.
We believe your feedback has led to improvements throughout the paper.
Kind Regards,
Ludger Heide
Reviewer 3 Report
Dear Authors,
The paper entitled “ Environmental Impact of Subsidy Concepts to Stimulate Car Sales in Germany” proposes a novel methodology to investigate the environmental impact of various car subsity concepts. The topic of the paper is of interest for the Journal Sustainability MDP. The paper can be pubblished after the following minor revisions:
- The abstract is very poor. It would be better to make it longer adding some numeric results and more considerations of the proposed analysis.
- The literature review could be extended by considering the following papers treating trends of CO2 emissions in Europe:
1)F. Famoso, R. Lanzafame, P. Monforte, PF Scandura. (2015). Analysis of the Covenant of Mayors Initiative in Sicily. Energy Procedia, 81, 482–492.
2)M. Lombardi, P. Pazienza, R. Rana. (2016). The EU environmental-energy policy for urban areas: The Covenant of Mayors, the ELENA program and the role of ESCos. Energy Policy, 93, 33-40.
3)C. Schenone, I. Delponte, I. Pittalunga. (2015). The preparation of the Sustainable Energy Action Plan as a city-level tool for sustainability: The case of Genoa. Journal of Renewable and Sustainable Energy, 7(3), article number 033126.
- I suggest to merge “Discussion” and “Conclusion in just one chapter about conclusions.
Kind Regards
Author Response
Dear Reviewer,
Thank you for your valuable feedback. We adjusted the abstract according to your suggestion. Thank you also for the interesting publications, we read all papers with great interest. To highlight the economic side of subsidy programs is of great importance. Unfortunately, the economic effects regarding a vehicle subsidy are outside of our scope, as we calculate the greenhouse gas emissions resulting from a fixed number of subsidized vehicles. However, as soon as researchers from economics deliver a reasonable prognosis of the economical effects, our model is able to calculate the resulting emissions for any number of subsidized vehicles.
In our opinion, our paper is improved by separating discussion and conclusion. In the discussion we discuss in detail all assumptions we used to model the subsidy concepts. In the conclusion we highlighted our main findings and concluded a recommendation. We prefer this structure unless there are specific reasons why both should be combined.
Kind Regards,
Ludger Heide